# Comparative inequalities in child dental caries across four countries: Examination of international birth cohorts and implications for oral health policy

**Sharon Goldfeld**[1,2,3]*, **Kate L. Francis**[2], **Elodie O'Connor**[1], **Johnny Ludvigsson**[4], **Tomas Faresjö**[5], **Beatrice Nikiema**[6,7], **Lise Gauvin**[6,8], **Junwen Yang-Huang**[9,10], **Yara Abu Awad**[11], **Jennifer J. McGrath**[11], **Jeremy D. Goldhaber-Fiebert**[12], **Åshild Faresjo**[5], **Hein Raat**[10], **Lea Kragt**[9,13], **Fiona K. Mensah**[2,3‡], **EPOCH Collaborative Group**[¶]

1 Centre for Community Child Health, Royal Children's Hospital, Melbourne, Victoria, Australia, 2 Murdoch Children's Research Institute, Royal Children's Hospital, Melbourne, Victoria, Australia, 3 Department of Paediatrics, The University of Melbourne, Melbourne, Victoria, Australia, 4 Crown Princess Victoria Children's Hospital and Div of Pediatrics, Dept of Biomedical and Clinical Sciences, Linköping University, Linköping, Sweden, 5 Dept of Health, Medicine and Caring Science, Linköping University, Linköping, Sweden, 6 Centre de Recherche du Centre Hospitalier de l'Université de Montréal (CRCHUM), Montréal, Québec, Canada, 7 Cree Board of Health and Social Services of James Bay [CBHSSJB], Department of Program Development and Support, Mistissini, Québec, Canada, 8 École de Santé Publique, Université de Montréal, Montréal, Québec, Canada, 9 The Generation R Study Group, Erasmus Medical Center, Rotterdam, The Netherlands, 10 Department of Public Health, Erasmus Medical Center, Rotterdam, The Netherlands, 11 PERFORM Centre & Department of Psychology, Montreal, Canada, 12 Stanford Health Policy, Centers for Health Policy and Primary Care and Outcomes Research, Department of Medicine, Stanford University, Stanford, California, United States of America, 13 Department of Oral & Maxillofacial Surgery, Erasmus Medical Centre, Rotterdam, The Netherlands

‡ FKM are joint senior authors on this work.
¶ Membership of the EPOCH Collaborative Group is provided in the Acknowledgments.
* Sharon.Goldfeld@rch.org.au

**Data Availability Statement:** Data underlying the results presented in this EPOCH study are available from the primary data sources. Data from LSAC is

## Abstract

Child dental caries (i.e., cavities) are a major preventable health problem in most high-income countries. The aim of this study was to compare the extent of inequalities in child dental caries across four high-income countries alongside their child oral health policies. Coordinated analyses of data were conducted across four prospective population-based birth cohorts (Australia, n = 4085, born 2004; Québec, Canada, n = 1253, born 1997; Rotterdam, the Netherlands, n = 6690, born 2002; Southeast Sweden, n = 7445, born 1997), which enabled a high degree of harmonization. Risk ratios (adjusted) and slope indexes of inequality were estimated to quantify social gradients in child dental caries according to maternal education and household income. Children in the least advantaged quintile for income were at greater risk of caries, compared to the most advantaged quintile: Australia: AdjRR = 1.18, 95%CI = 1.04–1.34; Québec: AdjRR = 1.69, 95%CI = 1.36–2.10; Rotterdam: AdjRR = 1.67, 95%CI = 1.36–2.04; Southeast Sweden: AdjRR = 1.37, 95%CI = 1.10–1.71). There was a higher risk of caries for children of mothers with the lowest level of education, compared to the highest: Australia: AdjRR = 1.18, 95%CI = 1.01–1.38; Southeast Sweden: AdjRR = 2.31, 95%CI = 1.81–2.96; Rotterdam: AdjRR = 1.98, 95%CI = 1.71–2.30; Québec:

available in a public, open-access repository (https://growingupinaustralia.gov.au/data-and-documentation). Data from GenR are available to request from (https://generationr.nl/researchers/); authors do not have permission to share their data. Data from ABIS are available to request from (http://www.abis-studien.se); authors do not have permission to share their data. For more information about the QLSCD cohort and data access, please see (https://www.jesuisjeserai.stat.gouv.qc.ca/informations_chercheurs/acces_an.html and https://www.maelstrom-research.org/study/qlscd); authors do not have permission to share their data. Other researchers can access these data in the same manner as the authors, as there were no special access privileges granted.

**Funding:** EPOCH was partly supported by Canadian Institutes of Health Research (J. McGrath OCO-79897, MOP-89886, MSH-95353; L. Séguin ROG-110537). Longitudinal Study of Australian Children (LSAC) was initiated and funded by Australian Government Department of Social Services, with additional funding from partner organizations Australian Institute of Family Studies (AIFS) and Australian Bureau of Statistics (ABS). This paper uses unit record data from Growing Up in Australia, the Longitudinal Study of Australian Children. The database of fluoride levels in water is maintained at the Australian Research Centre for Population Oral Health. The study was conducted in partnership with the Department of Social Services (DSS), the Australian Institute of Family Studies (AIFS) and the Australian Bureau of Statistics (ABS). The findings and views reported in this paper are those of the authors and should not be attributed to the DSS, the AIFS or the ABS. Generation R Study (GenR) was made possible by financial support from Erasmus Medical Center, Rotterdam; Erasmus University Rotterdam; Netherlands Organisation for Health Research and Development (ZonMw; additional grant received by V. Jaddoe, ZonMw 907.00303, 916.10159); Netherlands Organisation for Scientific Research (NWO); Ministry of Health, Welfare and Sport; and, Ministry of Youth and Families. GenR is conducted by Erasmus Medical Center in close collaboration with the School of Law and Faculty of Social Sciences of the Erasmus University Rotterdam, the Municipal Health Service Rotterdam area, Rotterdam, the Rotterdam Homecare Foundation, Rotterdam and the Stichting Trombosedienst & Artsenlaboratorium Rijnmond (STAR-MDC), Rotterdam; we gratefully acknowledge the contribution of children and parents, general practitioners, hospitals, midwives and pharmacies in Rotterdam. Québec Longitudinal Study of Child Development (QLSCD) 1996-2014 cohort was

AdjRR = 1.16, 95%CI = 0.98–1.37. The extent of inequalities varied in line with jurisdictional policies for provision of child oral health services and preventive public health measures. Clear gradients of social inequalities in child dental caries are evident in high-income countries. Policy related mechanisms may contribute to the differences in the extent of these inequalities. Lesser gradients in settings with combinations of universal dental coverage and/or fluoridation suggest these provisions may ameliorate inequalities through additional benefits for socio-economically disadvantaged groups of children.

## Introduction

Despite great improvements in the oral health of populations, dental caries (an infectious disease caused by certain types of bacteria [1]) remains a significant and preventable population health problem even in high-income countries [2, 3], and particularly for children [4]. This is highlighted by the recent WHO call for a global oral health strategy by 2022 involving major systems reforms placing equity and social justice at the core [5]. Dental caries is the most common disease of childhood [6] and affects up to 60–90% of school-aged children in most high-income countries [7–9]. Nearly 486 million children worldwide suffer from caries of primary teeth, posing a major global public health challenge [10].

Poor oral health can negatively impact the quality of life of children into adulthood. Major risk factors contributing to dental caries and its complications are the limited availability and accessibility of oral health services, poor oral hygiene behaviors (including inadequate tooth brushing, lack of flossing), adverse living conditions, and unhealthy lifestyles (including dietary patterns) [7, 11]. For example, frequent, excessive consumption of sugary food and beverages is thought to be a major cause of dental caries [7]. Poor oral health can impair a child's ability to eat, sleep, and socialize, which may result in ongoing adverse outcomes [1, 6, 12, 13]. These outcomes can include pain, high economic costs, poor nutrition, and impaired growth due to painful symptoms, school absenteeism, and lower quality of life [1, 14]. The health burden of dental caries is under-recognized, and socio-economically disadvantaged populations are disproportionately affected [15].

Children from low-income countries have a greater risk of caries and higher unmet treatment needs than those living in high-income countries [8]. The Global Burden of Disease 2017 study [2] showed that social inequalities exist in the distribution of oral health conditions such as dental caries. In 2017, the estimated prevalence of current dental decay (excluding missing or filled teeth) in deciduous teeth was 7.8% (532 million cases). While this burden lies primarily within low-income countries (265 million cases) a significant burden has been observed within high-income countries (41 million cases) [2]. Overall, the number of cases with current dental decay in deciduous teeth decreased in high-income countries between 1990 and 2017, while for low-income countries it has increased [2]. While the prevalence of dental caries in primary and permanent teeth varies for children in different continents [16], the overall high international prevalence calls for implementation of appropriate strategies to improve oral health and close equity gaps.

Poor oral health has a growing impact on vulnerable and marginalized populations of children. Families who are disadvantaged by low income and/or low education experience higher levels of dental caries, across both high-income and low-income countries [7, 17–20]. International reviews highlight the multifactorial aetiology of dental caries underpinned by the social determinants of health, but modulated by biological, social, economic, cultural, and

principally funded by l'Institut de la statistique du Québec through partnership with Fondation Lucie et André Chagnon, Ministère de l'Éducation et de l'Enseignement supérieur, Ministère de la Santé et des Services sociaux, Ministère de la Famille, GRIP Research Unit on Children's Psychosocial Maladjustment, QUALITY Cohort Collaborative Group, le Centre hospitalier universitaire Sainte-Justine, Institut de recherche Robert-Sauvé en santé et en securité au travail, l'Institut de recherche en santé publique de l'Université de Montréal, Centre de recherche du Centre hospitalier de l'Université de Montréal (CRCHUM), Fonds de recherche du Québec Santé (FRQS), Fonds de recherche du Québec Sociéte et culture (FRQSC), Social Sciences and Humanities Research Council (SSHRC), and Canadian Institutes of Health Research (MOP-123079, HDF-70335). The paper used unit record data from the QLSCD (ELDEQ – Enquête longitudinale des enfants du Québec). Data for the QLSCD were collected by the Institut de la Statistique du Québec, Direction des enquêtes longitudinales et sociales. ABIS (Alla Barn i Sydöstra Sverige; All Babies in Southeast Sweden) and this research were supported in part by the County Council of Ostergotland, Swedish Research Council (K2005-72X-11242-11A and K2008-69X-20826-01-4), the Swedish Child Diabetes Foundation (Barndiabetesfonden), Juvenile Diabetes Research Foundation, Wallenberg Foundation (K 98-99D-12813-01A), Medical Research Council of Southeast Sweden (FORSS), the Swedish Council for Working Life and Social Research (FAS2004–1775), and Ostgota Brandstodsbolag. Johnny Ludvisson founded the ABIS Cohort. Sharon Goldfeld is supported by Australian National Health and Medical Research Council (NHMRC) Practitioner Fellowship 1155290. Fiona Mensah was supported by NHMRC Career Development Fellowship 1111160. Research at the Murdoch Children's Research Institute is supported by the Victorian Government's Operational Infrastructure Program. The funders had no role in study design, data collection and analysis, decision to publish, or preparation of the manuscript.

**Competing interests:** The authors have declared that no competing interests exist.

environmental factors [21]. Recommended approaches to reduce prevalence include interventions that begin in the first year of life; evidence and risk-based management; and health care system financing that ensures the accessibility of preventive care [22].

The delivery of health promotion strategies including implementation of free dental care coverage for all children combined with public health measures, such as water fluoridation, have led to meaningful reductions in dental caries and social inequalities [21, 23–25]. Although the effectiveness of water fluoridation has been contested, in Australia water fluoridation is associated with reduced experience of caries, benefitting all socio-economic strata of the community [23]. International estimates suggest that water fluoridation reduces tooth decay by at least 25% in children and adolescents [24, 26], and that tooth decay in adults may be prevented by providing access to fluoridated water from an early age [24].

Child dental caries remains a significant and preventable chronic public health disease in high-income countries [27], with clear social inequalities and long term impacts on quality of life. International studies comparing cohorts from high-income countries with diverse health and social policies can facilitate our understanding of how variation in policy across jurisdictions impacts dental caries. Elucidating Pathways of Child Health Inequalities (EPOCH) is an international research project that aims to address knowledge gaps underlying childhood health disparities [28]. The EPOCH study, funded by the Canadian Institutes of Health Research (CIHR), was conceived and planned by researchers in the International Network for Research on Inequalities in Child Health (INRICH) [www.inrichnetwork.org]. This study utilized harmonized data from the EPOCH research project; specifically, four large, population-representative birth cohort studies in high-income countries with variable forms of universal health insurance and oral health policy: Australia, Québec (Canada), Rotterdam (the Netherlands), and Southeast Sweden (see S1 Table).

Comparisons across these high-income jurisdictions/countries can provide rich insight into child dental caries prevalence, and associated health policies and social inequalities. This research is both timely and necessary to advance potential future policy action. The aim of the current study was to investigate comparative social gradients and associated risk factors for child dental caries using longitudinal cohort data across four high-income countries. Observing differential childhood caries outcomes according to household income and maternal education would provide evidence to determine whether childhood dental caries was experienced equitably within high-income countries. Policy mechanisms that could explain any differences in the extent of inequalities, such as oral health care provision and water fluoridation, were then considered.

## Materials and methods

### Data sources

The EPOCH Collaborative Group investigates inequalities across a diversity of health outcomes up to age 10 years by comparing birth cohorts from high-income countries with diverse social policies. Coordinated analyses of health inequalities are being conducted across this series of cohorts using identical statistical methods and comparable variables. Analyses are underpinned by processes of harmonization to enable meaningful comparative interpretation of social gradients according to maternal income and education [28]. Concordia University Human Research Ethics Committee certified the ethical acceptability for EPOCH's secondary data use (#2011028). All original birth cohorts complied with the ethical standards of their relevant institutional and/or national committees and with the Helsinki Declaration of 1964, and its later amendments. Information summarizing what participation would involve was provided in both oral and written form.

For this study, data were drawn from four longitudinal cohorts within EPOCH, each including measures of dental caries: the national Longitudinal Study of Australian Children–Birth cohort (LSAC–B cohort) in Australia; and three regional studies: Québec Longitudinal Study of Child Development (QLSCD) in Québec, Canada; The Generation R Study in Rotterdam, the Netherlands; and, Alla Barn i Sydöstra Sverige (ABIS; English translation: All Babies in Southeast Sweden) in Southeast Sweden. Each study recruited women and children during pregnancy or infancy and prospectively followed up families. Further description of the cohort design, participants, survey weighting methods, ethics approvals and measures are provided in Table 1. All cohorts selected for inclusion included pertinent details regarding the nature/ administration and timing of dental caries measures or assessments, albeit study methods differed (e.g., caregiver report vs. dental exam photographs). Further, harmonization of these dental outcomes as well as coordinated analyses and harmonization of income and education classifications increased confidence that findings reflect robust jurisdictional differences. Measures included in this synthesis and the timing of administration are detailed as follows.

## Measures

**Outcome measure: Child dental caries.** Dental caries (yes/no) was measured across cohorts. In Australia (at age 8–9 years), Québec (at age 8–9 years), and Southeast Sweden (at age 5 years), the primary caregiver was asked to report on their child's history of dental caries including extractions and fillings within the study interview, while in Rotterdam (at age 6 years), direct observation of intraoral photographs was used. For each cohort, binary categories (yes/no) of dental caries were classified.

**Socio-economic position (SEP).** In line with the EPOCH foundational work undertaken to determine internationally comparable measures [28], maternal education level and household income were considered as indicators of socio-economic position, reflecting their wide applicability in epidemiological research, availability and comparability across cohorts. Maternal education was measured at baseline across all four cohorts; household income was measured within early childhood (Australia: at baseline (0–1 years); Québec: at baseline (0–1 years); Rotterdam: at age 6 years; and Southeast Sweden: at age 1–3 years). For each cohort, low/middle/high categories of maternal education were classified to provide comparative definitions across education systems. Low maternal education typically reflected low or incomplete secondary education; middle reflected completed secondary education, technical or vocational qualifications; and high reflected a university degree or higher (see Table 1 for full details).

Household income quintiles were classified within the participating cohorts, in Québec using gross income before tax (additionally adjusting regression models for household size); and in the other three cohorts using net income after tax and transfers had been accounted for.

**Oral health risk factors.** Children's consumption of sugary food and sugary drinks (classified within the cohorts as less than daily/once a day/more than once a day) were measured across cohorts (at ages 8–9 years in Australia, 10 years in Québec, 6 years in Rotterdam, and 5 years in Southeast Sweden). Jurisdictions varied in whether and how tooth brushing was measured, so it was not included due to being unable to derive a consistent indicator. Ideally, whether oral health services were provided for preventive care would have also been included. It was also unable to be ascertained whether families sought care for preventive reasons or in response to caries experienced by the child. Oral health service provision was thus explored at a policy level by jurisdiction. Similarly, naturally occurring water fluoride and policies for fluoridation were explored at the jurisdiction level.

**Table 1. Cohort descriptions and variable definitions.**

| | Australia (LSAC) | Quebec, Canada (QLSCD) | Rotterdam, the Netherlands (Gen R) | Southeast Sweden (ABIS) |
|---|---|---|---|---|
| **Cohort description** | LSAC is a nationally representative sample of two cohorts of Australian children—the birth cohort (B-cohort) of 5107 infants, and the kindergarten cohort (K-cohort) of 4983 4-year-olds—each of which commenced in May 2004 [29]. Follow-ups were conducted every two years. | QLSCD follows a representative sample of 2120 singleton live births, born in 1997–1998 to mothers living in Quebec. Annual follow-ups were conducted until 8 years of age, with approximately biennial follow-ups after this time. | Generation R Study is a population-based prospective cohort study. The 9778 mothers enrolled in the study gave birth to 9749 live born children [30]. | ABIS is a prospective cohort study of 17,055 children (78.6% of all babies born in Southeast Sweden between October 1997 and September 1999). Follow-ups were conducted at approximately 1 year, 3 years, 5 years, 8 years, and 10–12 years of age. |
| **Recruitment & exclusion criteria (if applicable)** | A national sample that was broadly representative of all Australian children except those living in remote areas was recruited using two-stage random sampling design: (1) random selection of 10% of postcodes, stratified by state and urban/rural locations), (2) random selection of in-age children within those postcodes from Medicare (universal healthcare) database [31]. The sampling excluded very remote postcodes and postcodes with <20 children (n = 874 postcodes, 3.2% of population). The LSAC design and sampling methodology is documented elsewhere [29, 31]. | All singleton live births, born in 1997 from mothers living in Quebec, except in First Nation's territories (except for those born to mothers living in Northern Quebec, the Cree and Inuit territories or on Indian reserves) Additional exclusion criteria included undisclosed sex, unknown gestational age, very premature birth (<24 weeks) or very post-term birth (> 42 weeks) [32, 33]. | Midwifes and obstetricians invited all pregnant women under their care with an expected delivery date between April 2002 and January 2006, living in Rotterdam in the Netherlands at time of delivery, to participate.[5] Details on the study design and participant inclusion procedure has been published previously [34]. | All children born October 1, 1997 to September 30, 1999 in a defined region in southeast of Sweden were invited to participate. Details on the study design and population are detailed elsewhere [35, 36]. |
| **Study years** | 2004 to present | 1997 to present | 2002 to 2006 | 1997 to 1999 |
| **Waves** | Baseline (Wave 1): Birth-1yr; Wave 2: 2–3 yrs; Wave 3: 4–5 yrs; Wave 4: 6–7 yrs; Wave 5: 8–9 yrs; Wave 6: 10–11 yrs | Baseline (Wave 1): 6 mths; Wave 2: 1.5 yrs; Wave 3: 2.5 yrs; Wave 4: 3.5 yrs; Wave 5: 4 yrs; Wave 6: 5 yrs; Wave 7: 6 yrs; Wave 8: 7 yrs; Wave 9: 8 yrs; Wave 10: 10 yrs | Baseline (Wave 1): Birth-4yrs ("Preschool Period": 2 mths, 6 mths, 1 yr, 1.5 yrs, 2 yrs, 3 yrs, 4 yrs); Wave 2: 5–6 yrs; Wave 3: 9–10 yrs | Baseline (Sweep 1): Birth; Sweep 2: 1 yr; Sweep 3: 2.5 yrs; Sweep 4: 5 yrs; Sweep 5: 8 yrs; Sweep 6: 10–12 yrs |
| **Participants** | | | | |
| At baseline | n = 5,107 | n = 2,120 | n = 9,749 | n = 17,055 |
| In childhood | 8–9 yrs (Wave 5): n = 4,085 | 8 yrs (Wave 9): n = 1,451 | 6 yrs (Wave 3): n = 8,305 | 5 yrs (Sweep 4): n = 7,445 |
| Retention rate | 80% | 68% | 85% | 44% |
| **Data used** | This paper uses data from the B-cohort from ages 0–1 to 8–9 years of age (n = 4085). | This paper uses data from birth to 8 years of age (n = 1253). | From all included children, consent for follow-up was available for 8305 children at aged 6 years. This paper uses oral health data from 6690 children who visited the research center at 6 years of age [37]. | From all included children. This paper uses oral health data from 7445 children who visited the research center at 5 years of age. |
| **Survey design** | Survey sample weights recalibrated the data so the sample participating at age 8–9 years was more representative of the original target sample of the population; these weights take account of the initial sample selection and initial non-response as well as each stage of participation up to age 8–9 years [38, 39]. | Weights were used to adjust for non-response and sampling probabilities such that the population at follow-up represented the population sampled at baseline. | Stabilized inverse probability weights were built in order to adjust for differential loss to follow-up using methodology proposed by Hernan and Robins [40]. | To adjust for bias due to non-response, stabilized weights were estimated using the probability of being lost to follow up conditional on maternal education and income in the denominator and the joint probability of both in the numerator. |

*(Continued)*

**Table 1.** (Continued)

|  |  | Australia (LSAC) | Quebec, Canada (QLSCD) | Rotterdam, the Netherlands (Gen R) | Southeast Sweden (ABIS) |
|---|---|---|---|---|---|
| **Ethics approval** |  | The LSAC methodology was approved by the Australian Institute of Family Studies Human Research Ethics Review Board. | Ethical approval for data collection was obtained from the ethics boards of the Institut de la statistique du Québec, the Centre Hospitalier Universitaire (CHU) Sainte Justine, and the Faculty of Medicine of Université de Montréal [32, 33]. Approval was also obtained from the Comité d'éthique à la recherche du Centre Hospitalier de l'Université de Montréal (CHUM) for EPOCH project analyses. | The study was conducted in accordance with the guidelines proposed in the World Medical Association Declaration of Helsinki and was approved by the Medical Ethical Committee at Erasmus MC, University Medical Center Rotterdam. | The study was approved by The Research Ethics Committé, Linköping University (Dnr LiU 287–96) and Lund University (Dnr 83–97 and Dnr 03–092), with access to national registers (Dnr 03–513). ABIS is connected to the National Registry of Diagnosis and the National Registry of Drug prescriptions. |
| **Participant consent** |  | Explicit informed written consent was obtained for all participants (parents, guardians) across all cohorts. | | | |
| **Outcome measure** |  |  |  |  |  |
|  | Dental caries | Item: Parent report: Has the study child **ever** had any of the following problems with his/her teeth:<br>(1) Cavities or dental decay?<br>(2) Tooth or teeth filled because of dental decay?<br>(3) Teeth pulled because of dental decay?<br>Who conducted examination: N/A<br>Parent-report: Computer assisted interview<br>Scoring: A "yes" to any of the 3 questions is a positive response<br>Age measured: 8–9 years | Item: Parent report: Untreated cavities; treated for cavities; extracted because of cavities (up to 8/9 years)<br>Who conducted examination: N/A<br>Parent-report: Interview<br>Scoring: A "yes" to any of the questions is a positive response<br>Age measured: Mean age 8/9 years | Item: Direct observation of intraoral photographs: Count of the number of decayed, missing or filled teeth due to cavities<br>Who conducted examination: The presence of dental caries on intraoral photographs was scored by using the decayed, missing, and filled teeth index (dmft index). The intraobserver reliability was (K = 0.98) and inter-observer reliability was (K = 0.89), both indicating an almost perfect agreement [18]. The intraoral photographs pictures were taken with either the Poscam USB intraoral (Digital Leader PointNix) or Sopro 717 (Acteon) autofocus camera. Both cameras had a resolution of 640 × 480 pixels and a minimal scene illumination of 1.4 and 30 lx [18].<br>Parent-report: N/A<br>Scoring: Any value greater than 0 is a positive response<br>Age measured: 6 years | Item: Parent report: Has the child any cavities?<br>Responses could be number from 0 to 8<br>Who conducted examination: N/A<br>Parent-report: Questionnaire<br>Scoring: Any value greater than 0 is a positive response<br>Age measured: 5 years |
| **Demographic variables** |  |  |  |  |  |
|  | Sex | Age measured: Baseline (0–1 years) | Age measured: Baseline (0–1 years) | Age measured: Baseline (at birth) | Age measured: Baseline (0–1 years) |
|  | Geographic area | Item: Using the accessibility/remoteness index of Australia[a] children's local areas were categorized as a major city, inner regional, outer regional, remote, or very remote area.<br>Age measured: Baseline (0–1 years) | Item: Using geographical indicators developed by Statistics Canada, families' residential area were categorized as being located either in a census metropolitan area, a census agglomeration, or a rural area. Children living in census agglomerations or rural areas were considered as being in "remote" areas.<br>Age measured: 8–9 years | Not applicable: The study only included participants living in Rotterdam at baseline (a city) | Item: Categorized as city/urban or rural/remote<br>Age measured: Baseline (0–1 years) |

(*Continued*)

**Table 1.** (Continued)

| | Australia (LSAC) | Quebec, Canada (QLSCD) | Rotterdam, the Netherlands (Gen R) | Southeast Sweden (ABIS) |
|---|---|---|---|---|
| **Socio-economic position** | | | | |
| Maternal education[b] | Item: Classified based on the International Standard Classification of Education.[c] Level of education was classified as: low (Year 10 or equivalent; Year 9 or equivalent; Year 8 or below; never attended school; still at school), middle (Year 12 or equivalent; Year 11 or equivalent; advanced diploma/diploma; certificate; trade certificate/apprenticeship), and high (postgraduate degree; graduate diploma/certificate; bachelor degree). Age measured: Baseline (0–1 years) | Item: Classified based on the International Standard Classification of Education.[c] Level of education was classified as: low (Secondary School Diploma (SSD); Year 11 or lower or equivalent; diploma of vocational studies), middle (SSD & college studies diploma or certificate), and high (University certificate or diploma). Age measured: Baseline (0–1 years) | Item: Classified based on the International Standard Classification of Education.[c] Level of education was classified as: low (no education, primary school, lower vocational training, intermediate general school, or four years or less general secondary school), middle (more than four years general secondary school, intermediate vocational training, or first year of higher vocational training), and high (higher vocational training, university or PhD degree). Age measured: Baseline (during pregnancy or at child 0–1 years) | Item: Classified based on the International Standard Classification of Education.[c] Level of education was classified as: low (Year 9 or equivalent), middle (Year 12 or equivalent), and high (over year 12 postgraduate degree university). Age measured: Baseline (0–1 years) |
| Household income quintile | Item: Families' combined weekly household income Age measured: Baseline (0–1 years) | Item: Household income before tax, during the 12 months prior to maternity leave Age measured: Baseline (0–1 years) | Item: Net household income per month Age measured: 6 years | Item: Household income after taxations and transfers Age measured: 1–2 years |
| **Oral health risk factors** | | | | |
| Consumption of sugary foods | Item: Child ate cakes, candy chocolate bar in previous 24hrs Age measured: 8–9 years | Item: Number of times per week or day child has eaten as a snack: cookies pastries, granola bar and candies Age measured: Mean age 8–9 years | Item: How often eaten a high calorific snack? Age measured: 6 years | Item: How often are your child eating cakes, Danish pastry, biscuits, eating chocolate candy or cake? Age measured: 5 years |
| Consumption of sugary drinks | Item: Child consumed sugary drinks in previous 24hrs Age measured: 8–9 years | Item: Number of times per week or day child has eaten as a snack: fruit drinks, soft drinks Age measured: Mean age 8–9 years | Item: How many sweet drinks on average weekday/weekend? Age measured: 6 years | Item: How often are your child drinking sweetened soft drinks/juices Age measured: 5 years |

[a]Department of Health and Aged Care. Measuring remoteness: Accessibility/Remoteness Index of Australia (ARIA). Canberra, Australia: Commonwealth of Australia; 2001.

[b]There are differences in the education system across countries.

[c]United Nations Educational Scientific and Cultural Organization. International Standard Classification of Education: ISCED 1997. UNESCO; 1997.

[d]Household size (number of household members) is included as a control variable for the QLSCD study because the household income quintile uses gross income before tax.

**Demographic variables.** For each cohort, children's age and sex (classified as male/female) were recorded, along with geographic area (city or urban/rural or remote).

## Statistical analysis

**Child dental caries by cohort.** Rates of the child having ever experienced dental caries by the applicable age at follow up were estimated for each of the cohorts.

**Within cohort social gradients.** Relative risk ratios were estimated to quantify the association between maternal education, income, and childhood dental caries. These estimates were adjusted to account for: sex, geographic area, and sugary food and drink consumption (as available for each cohort). Risk ratios were estimated within each population using a generalized linear model with log link function and robust variance estimation [41].

The Slope Index of Inequality (SII) was estimated for each cohort, representing the absolute difference in caries prevalence that would be expected between children in the most and least advantaged households (according to income and maternal education respectively) [42]. The SII is a regression-based index estimating the magnitude of inequalities in health. Differences between the least and the most advantaged groups were estimated for each cohort using linear regression of the weighted prevalence estimates for each of the income quintiles/three levels of maternal education with the mid points of the cumulative fractions of the socio-economic groups as the predictors.

Australian and Southeast Sweden data were analyzed using Stata version 15. Québec data were analyzed using SPSS version 25. Rotterdam data were analyzed using R version 3.6.1.

**Policy and service context.** The oral health care policy and service systems were described for each jurisdiction to detail local contexts, noting public funding, out of pocket expenses, access to dental services, and water fluoridation policies (S1 Table).

## Results

### Cohort characteristics

Cohort characteristics such as participant retention rates are detailed in Table 1. Cohorts had a similar balance of males and females (Table 2). The Rotterdam cohort recruited an urban population, while the other three cohorts included both urban and non-urban families at recruitment (during pregnancy or infancy).

**Child dental caries.** By age 8/9 years, over half (55%) of Québec children had experienced caries (parent-reported untreated cavities, treated cavities and extracted teeth; Table 2). In Australia, the percentage of children with caries was 41% by age 8/9 years (parent-reported decayed, missing, or filled teeth). In Rotterdam, dental caries by age 6 years were identified for 25% of children (intraoral photographs to assess decayed, missing or filled teeth due to cavities) but 20% of the children were not assessed, so the prevalence might be higher (31.5% amongst children assessed). In Southeast Sweden, caries were lowest at 12% for children aged 5 years (parent-reported cavities). Direct comparisons of prevalence are cautioned given differences in age and assessment of caries.

### Risks for child dental caries within each jurisdiction

**Inequalities by income and maternal education.** Fig 1 illustrates adjusted risk ratios examining association between income, maternal education and caries for each cohort. Adjusted risk ratios for Australia (AdjRR = 1.18, 95%CI = 1.01–1.38), Southeast Sweden (AdjRR = 2.31, 95%CI = 1.81–2.96), Rotterdam (AdjRR = 1.98, 95%CI = 1.71–2.30), and Québec (AdjRR = 1.16, 95%CI = 0.98–1.37) indicate higher risk of caries for children of mothers with the lowest level of education (Table 3).

Examining income, those in the least advantaged quintile were at greater risk of caries for all cohorts (compared to the most advantaged quintile: Australia: AdjRR = 1.18, 95% CI = 1.04–1.34; Québec: AdjRR = 1.69, 95%CI = 1.36–2.10; Rotterdam: AdjRR = 1.67, 95% CI = 1.36–2.04; Southeast Sweden: AdjRR = 1.37, 95%CI = 1.10–1.71). Income gradients within Rotterdam and Québec reflected higher risk according to each quintile of disadvantage.

**Oral health risk factors.** Consumption of sugary drinks or foods varied substantially across the four cohorts (Table 2). Accordingly the associations with dental caries also varied (Table 3). In Australia, child dental caries was associated with high sugary beverage consumption (more than once a day vs less than daily, AdjRR = 1.19, 95%CI = 1.05–1.34). In Québec, child dental caries was associated with sugary food consumption (once a day, AdjRR = 1.25, 95%CI = 1.09–1.44; more than once a day AdjRR = 1.32, 95%CI = 1.16–1.49). In Rotterdam

**Table 2. Participant details for each cohort.**

| | | Australia (LSAC) | | Québec, Canada (QLSCD) | | Rotterdam, the Netherlands (Gen R) | | Southeast Sweden (ABIS) | |
|---|---|---|---|---|---|---|---|---|---|
| | | N | % | N | % | N | % | N | % |
| **Dental caries** | | | | | | | | | |
| | Yes | 1676 | 41.0 | 693 | 55.3 | 1678 | 25.1 | 891 | 12.0 |
| | No | 2375 | 58.1 | 560 | 44.7 | 3649 | 54.5 | 6509 | 87.4 |
| | *Missing* | 34 | 0.8 | 0 | 0 | 1363 | 20.4 | 45 | 0.6 |
| **Sex** | | | | | | | | | |
| | Male | 2096 | 51.3 | 606 | 48.3 | 3352 | 50.1 | 3885 | 52.2 |
| | Female | 1989 | 48.7 | 647 | 51.7 | 3338 | 49.9 | 3558 | 47.8 |
| | *Missing* | 0 | 0 | 0 | 0 | 0 | 0 | 0 | 0 |
| **Geographic area** | | | | | | | | | |
| | City/urban[a] | 2584 | 63.3 | 805 | 64.3 | 6690 | 100.0 | 5290 | 71.0 |
| | Rural/remote | 1500 | 36.7 | 443 | 35.3 | 0 | 0 | 1807 | 24.3 |
| | *Missing* | 1 | 0.0 | 5 | 0.4 | 0 | 0 | 348 | 4.7 |
| **Maternal education** | | | | | | | | | |
| | High | 1481 | 36.3 | 364 | 29.1 | 2846 | 42.5 | 2339 | 31.4 |
| | Middle | 2202 | 53.9 | 531 | 42.4 | 1885 | 28.2 | 4396 | 59.0 |
| | Low | 400 | 9.8 | 357 | 28.5 | 1345 | 20.1 | 633 | 8.5 |
| | *Missing* | 2 | 0 | 0 | 0 | 614 | 9.2 | 77 | 1.0 |
| **Household income quintile** | | | | | | | | | |
| | Quintile 1 (highest) | 911 | 22.3 | 211 | 16.9 | 1178 | 17.6 | 1253 | 16.8 |
| | Quintile 2 | 887 | 21.7 | 216 | 17.2 | 612 | 9.1 | 1456 | 19.6 |
| | Quintile 3 | 835 | 20.4 | 268 | 21.4 | 1444 | 21.6 | 1496 | 20.1 |
| | Quintile 4 | 806 | 19.7 | 254 | 20.3 | 1208 | 18.1 | 1558 | 20.9 |
| | Quintile 5 (lowest) | 646 | 15.8 | 226 | 18.0 | 910 | 13.6 | 1606 | 21.6 |
| | *Missing* | 0 | 0 | 77 | 6.2 | 1338 | 20.0 | 76 | 1.0 |
| **Consumption of sugary foods** | | | | | | | | | |
| | Less than daily | 1004 | 24.6 | 769 | 61.4 | 1509 | 22.6 | 7122 | 95.7 |
| | Once a day | 2032 | 49.7 | 208 | 16.6 | 1270 | 19.0 | 174 | 2.3 |
| | More than once a day[b] | 1003 | 24.6 | 274 | 21.9 | 2824 | 42.2 | 0 | 0 |
| | *Missing* | 46 | 1.1 | 1 | 0.1 | 1087 | 16.2 | 149 | 2.0 |
| **Consumption of sugary drinks** | | | | | | | | | |
| | Less than daily | 2726 | 66.7 | 1063 | 84.8 | 686 | 10.3 | 5969 | 80.2 |
| | Once a day | 936 | 22.9 | 94 | 7.5 | 519 | 7.8 | 1312 | 17.6 |
| | More than once a day[b] | 382 | 9.4 | 95 | 7.6 | 4370 | 65.3 | 0 | 0 |
| | *Missing* | 41 | 1.0 | 1 | 0.1 | 1115 | 16.7 | 164 | 2.2 |

[a]Gen R only included participants living in the city of Rotterdam.
[b]ABIS did not include the option "more than once a day".

and Southeast Sweden, there was little evidence to suggest that consumption of sugary foods or drinks was associated with child dental caries.

**Geographic risk factors.** In Australia, child dental caries was associated with living in rural or remote areas compared to urban areas (AdjRR = 1.22, 95%CI = 1.12–1.33). Differences in caries risk according to geographic area were less evident for children in Québec and Southeast Sweden (children in Rotterdam were all living in urban areas at birth).

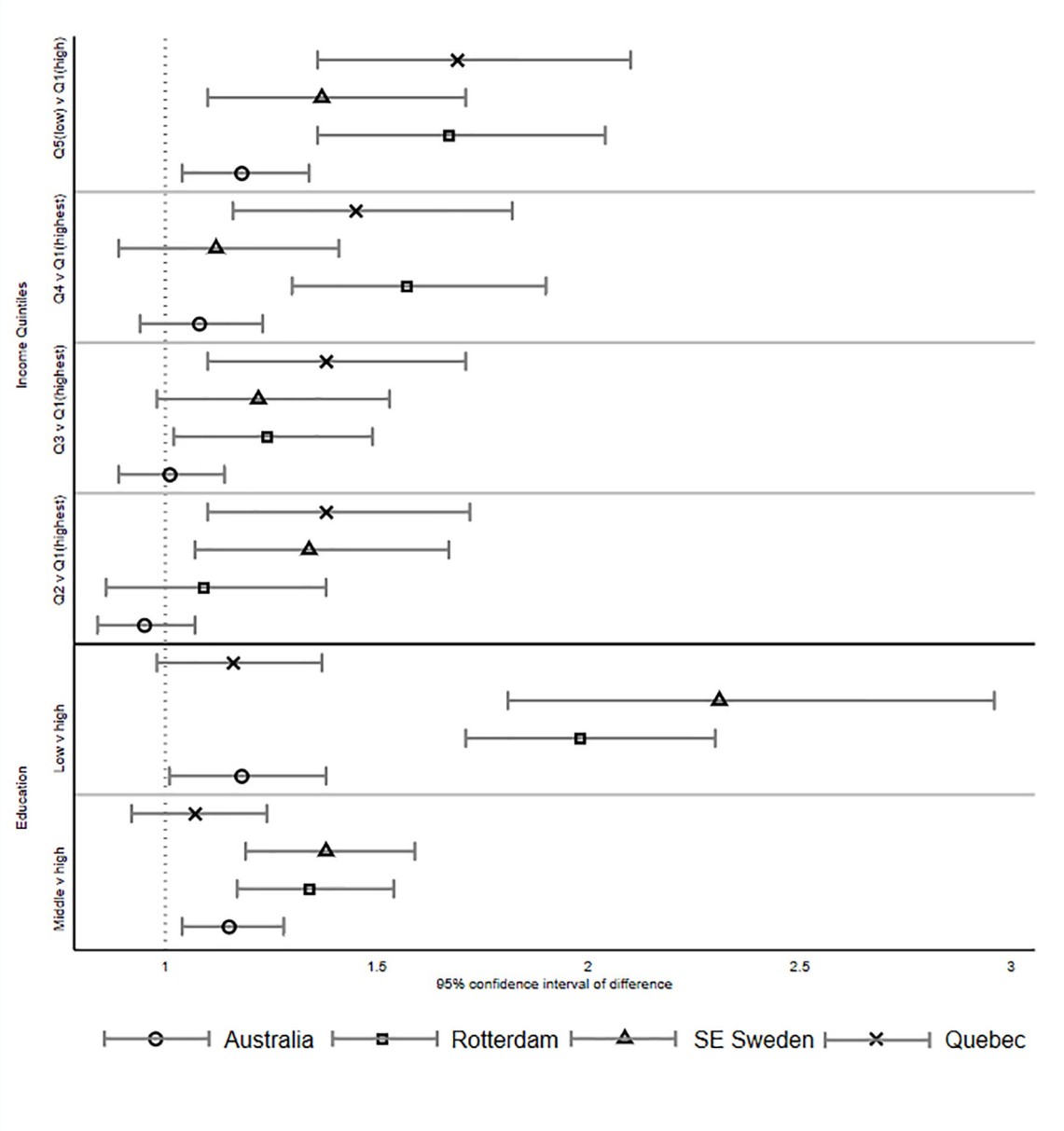

**Fig 1. Forest plots illustrating inequalities in child dental caries by income and maternal education.** Forest plots showing the association between (1) household income and child dental caries comparing the highest income to the each of the four lower categories; (2) mother's education and child dental caries with the highest education category compared to the middle level and lowest level.

## Slope Index of Inequality (SII)

SII by maternal education and income are illustrated for each cohort in Fig 2. For maternal education, the strongest gradient was in Rotterdam (SII = -45), estimating a 45% difference in caries prevalence when comparing the children whose mothers had the lowest level of education to the highest. Shallower gradients for Québec, Southeast Sweden, and Australia (SII = -30, -16, -15, respectively) reflected less difference in prevalence of child dental caries according to maternal education. For income quintiles, the gradients for Québec and Rotterdam

**Table 3. Adjusted risk ratios for child dental caries.**

| | | Australia (LSAC) | | | Québec, Canada (QLSCD) | | | Rotterdam, the Netherlands (Gen R) | | | Southeast Sweden (ABIS) | | |
|---|---|---|---|---|---|---|---|---|---|---|---|---|---|
| | | Adj RR[a] | [95% CI] | | Adj RR[a] | [95% CI] | | Adj RR[a] | [95% CI] | | Adj RR[a] | [95% CI] | |
| | | | Lower | Upper | | Lower | Upper | | Lower | Upper | | Lower | Upper |
| **Sex** | | | | | | | | | | | | | |
| | Male | ref | | | ref | | | ref | | | ref | | |
| | Female | 1.02 | 0.94 | 1.11 | 1.00 | 0.90 | 1.11 | 1.00 | 0.90 | 1.11 | 1.09 | 0.96 | 1.24 |
| **Geographic area** | | | | | | | | | | | | | |
| | City/urban | ref | | | ref | | | ref | | | ref | | |
| | Rural/remote[b] | 1.22 | 1.12 | 1.33 | 1.09 | 0.98 | 1.22 | | | | 1.00 | 0.85 | 1.15 |
| **Maternal education** | | | | | | | | | | | | | |
| | High | ref | | | ref | | | ref | | | ref | | |
| | Middle | 1.15 | 1.04 | 1.28 | 1.07 | 0.92 | 1.24 | 1.34 | 1.17 | 1.54 | 1.38 | 1.19 | 1.59 |
| | Low | 1.18 | 1.01 | 1.38 | 1.16 | 0.98 | 1.37 | 1.98 | 1.71 | 2.30 | 2.31 | 1.81 | 2.96 |
| **Household income quintile** | | | | | | | | | | | | | |
| | Quintile 1 (highest) | ref | | | ref | | | ref | | | ref | | |
| | Quintile 2 | 0.95 | 0.84 | 1.07 | 1.38 | 1.10 | 1.72 | 1.09 | 0.86 | 1.38 | 1.34 | 1.07 | 1.67 |
| | Quintile 3 | 1.01 | 0.89 | 1.14 | 1.38 | 1.10 | 1.71 | 1.24 | 1.02 | 1.49 | 1.22 | 0.98 | 1.53 |
| | Quintile 4 | 1.08 | 0.94 | 1.23 | 1.45 | 1.16 | 1.82 | 1.57 | 1.30 | 1.90 | 1.12 | 0.89 | 1.41 |
| | Quintile 5 (lowest) | 1.18 | 1.04 | 1.34 | 1.69 | 1.36 | 2.10 | 1.67 | 1.36 | 2.04 | 1.37 | 1.10 | 1.71 |
| **Consumption of sugary foods** | | | | | | | | | | | | | |
| | Less than daily | ref | | | ref | | | ref | | | ref | | |
| | Once a day | 0.95 | 0.86 | 1.05 | 1.25 | 1.09 | 1.44 | 0.95 | 0.82 | 1.10 | 1.25 | 0.84 | 1.85 |
| | More than once a day[c] | 1.08 | 0.96 | 1.21 | 1.32 | 1.16 | 1.49 | 1.04 | 0.91 | 1.18 | | | |
| **Consumption of sugary drinks** | | | | | | | | | | | | | |
| | Less than daily | ref | | | ref | | | ref | | | ref | | |
| | Once a day | 1.02 | 0.93 | 1.12 | 0.96 | 0.77 | 1.19 | 1.01 | 0.81 | 1.27 | 0.98 | 0.83 | 1.17 |
| | More than once a day[c] | 1.19 | 1.05 | 1.34 | 1.06 | 0.88 | 1.28 | 0.95 | 0.81 | 1.12 | | | |
| **Household size** (mean centered)[d] | | - | - | - | 1.05 | 1.00 | 1.10 | | | | | | |

[a]Caries risk adjusted for all applicable measures as detailed in the table above, including sex, geographic area, maternal education, income quintile, and sugary food and drink consumption (plus household size for QLSCD see note [d]).

[b]Gen R only included participants living in the city of Rotterdam.

[c]ABIS did not include the option "more than once a day".

[d]Household size (number of household members) is included as a control variable for the QLSCD study because the household income quintile uses gross income before tax.

were similar (-36 and -34); Australia had a shallower gradient (-16); while the Southeast Sweden line was close to flat (-2). The SII estimates absolute unadjusted differences with the interpretation that, in the absence of confounding, if all families were as wealthy as the richest family, the prevalence of caries would be 2% lower for Southeast Sweden, 16% lower for Australia, 34% lower for Rotterdam, and 36% lower for Québec.

## Discussion

The health burden of child dental caries within high-income countries is under-recognized and preventable, disproportionately affecting socio-economically disadvantaged families. This study aimed to develop policy-relevant evidence, specifically targeting inequalities, through

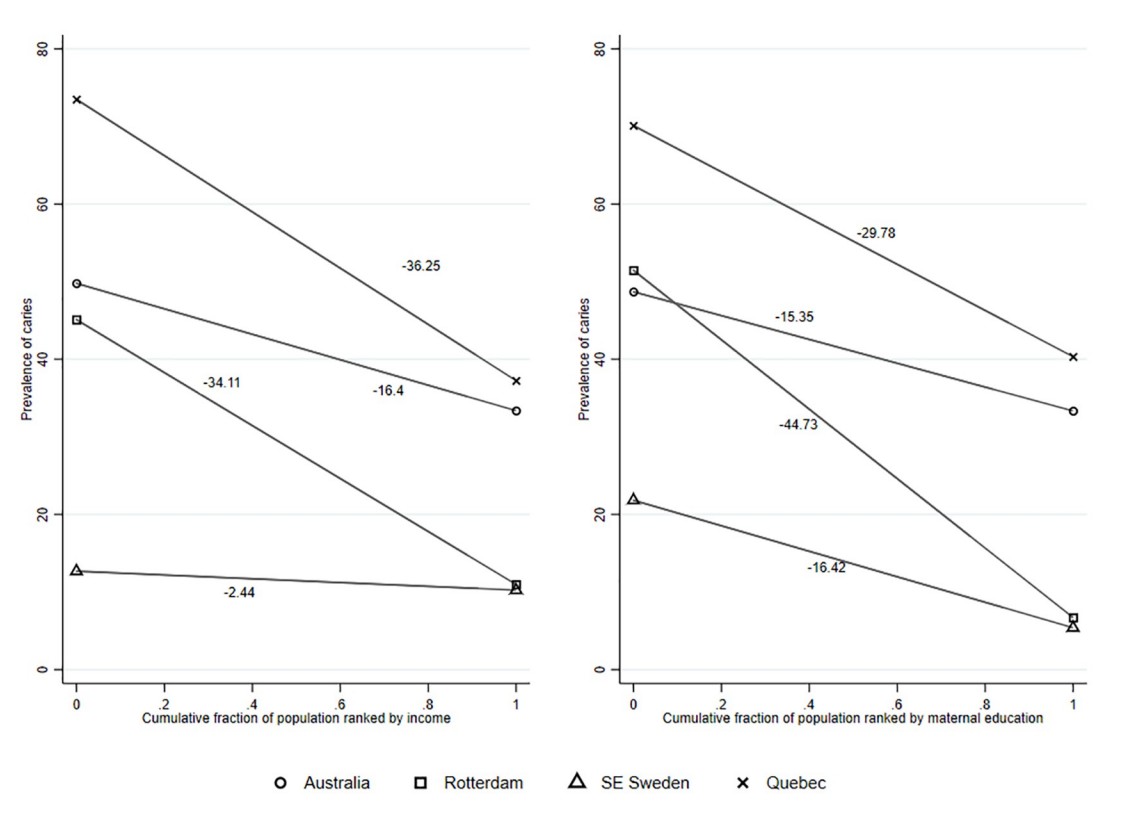

**Fig 2. Slope Index of Inequality (SII) in child dental caries by income and education.**

the comparison of rates of child dental caries and the extent of inequalities across high-income countries (Australia, Québec (Canada), Rotterdam (the Netherlands), and Southeast Sweden). Using data from large, representative birth cohorts, this study found variability in prevalence and distribution of child dental inequalities across these jurisdictions. Overall, the identified prevalence of caries varied from 12% to 55% of the population, noting the differences in ages (i.e., 8–9 years for Québec versus 5 years for Southeast Sweden).

In all countries, the current study found that children living in families with lower maternal education or household income were more likely to experience dental caries. Southeast Sweden and Rotterdam had greater inequality related to maternal education than Australia and Québec. Inequalities according to income were greatest in Québec and Rotterdam. Taking into account underlying population income distribution, the SII for Rotterdam and Québec were similarly steep (-34 and -36), than Australia's more moderate SII (-16). The SII for Southeast Sweden was close to flat (-2), which may reflect the social system with transfer of money to families with low income, in combination with completely free childhood dental care (including preventive care) and natural fluoridation of the drinking water. In contrast, the Netherlands similarly has free dental care but the gradient was still evident. These estimates suggest potential absolute reductions in dental caries of between 2% and 36% if inequalities were ameliorated and all children held the same advantage as the richest families in each jurisdiction.

Findings from the present study were consistent with previous research demonstrating social inequalities in child dental caries within high-income countries. In Australia (also using

the LSAC cohort), the odds of having dental caries were highest for children living in families with low socio-economic position (SEP: a composite of family income, education and occupation), and in areas with non-fluoridated water supply [43]. Low maternal education has also been shown to be associated with child dental caries in a longitudinal study of Indigenous Australian children [44]. In the Netherlands (also using the Generation R cohort), low maternal education was associated with higher odds of dental caries [18]. In Sweden, low maternal education and low family income were associated with higher rates of child dental caries [45].

The different levels of inequality in dental caries across the cohorts are likely to reflect substantive differences in child oral health policy across jurisdictions. Drawing on the oral health and service policies detailed in S1 Table, one potential reason may be due in part to the protective effect of water fluoridation. Australian research has shown that fluoridation of the water supply may have a beneficial impact in preventing dental caries, to some extent ameliorating the social gradient [24, 43]. This evidence is limited, with a Cochrane review reporting insufficient evidence to support this claim due to the general absence of causal evidence [46]. Australia mostly has very low levels of naturally occurring fluoride; yet, community water fluoridation means that around 89% of Australians have access to fluoridated drinking water [24]. Sweden's drinking water only contains a natural variation of fluoride; however, fluoride levels do vary within municipalities [47]. While the current study was unable to ascertain this relationship, it is plausible that the artificially fluoridated drinking water in Australia and natural fluoridation in Sweden contributes to the lower levels of inequality demonstrated in these two countries. Previous Australian research has shown that fluoridation of the water supply seems to ameliorate (although not eliminate) inequalities [43]. In Québec, fluoridation of water is not widespread; less than 3% of the population has access to fluoridated water [48]. This has been identified as one likely reason why dental caries is far more prevalent than in neighboring provinces like Ontario (over 70% fluoridated water) [49]. In the Netherlands, fluoride is not added to the drinking water; the natural concentration of fluoride varies between 0.05 and 0.25 mg/L.

Further differences for socio-economically disadvantaged families may relate to the inclusion of dental care in universal health coverage. Universal coverage reduces barriers to care which addresses caries through prevention and prompt treatment [2]. In the current study there was evidence of considerable variability in oral health services (S1 Table). For example, in Australia and Québec, the publicly funded health system provides limited oral health coverage, potentially resulting in large out of pocket expenses. In Québec, children aged under 10 years have access to free dental services, but their parents do not have this coverage, which may impede service use [50]. In the Netherlands, the costs for basic dental care for children (until 18 years) are reimbursed via the obligatory (by law) 'basic health insurance' [18]. In Sweden, however, oral health is free of charge from birth to 23 years of age with an expectation and encouragement for preventive dental care.

## Strengths and limitations

A major strength of the current study was the data from four large, population representative cohort studies which enabled comparisons from jurisdictions providing rich, valuable information on child dental caries and social inequalities. However, this synthesis also posed some challenges, as the independent cohorts were not designed together. Some measures, such as maternal education, were captured in nearly identical ways and at similar time-points across all four cohorts. Others, such as child dental caries, were measured using different assessment methods at different ages across cohorts (see Table 1 for more detail).

In Australia, Québec, and Southeast Sweden, child dental caries was measured using parent-report within the survey interview, while the Rotterdam cohort used direct observation of intraoral photographs. Validation of parent-reported single-item indicators of child dental caries has recognized their potential underestimation of dental caries [51]. Using dental clinical examinations or dental records to validate parent-reported dental caries would be ideal in future studies [17]. Different criteria for identifying child dental caries means that caution should be held in interpreting differences in prevalence between jurisdictions [22]. However, the coordinated analysis enabled a high degree of harmonization, increasing confidence that findings reflect actual population differences.

Policy modifiable oral health factors, such as water fluoride level and use of dental services, are known to impact inequalities in rates of child dental caries in Australia [43]. However, data were not available to examine these relationships internationally. Future research could endeavor to examine the role of these factors on inequalities in rates of child dental caries across countries.

Finally, each cohort has its own limitations that should be recognized including population groups for whom the current findings may not apply. In the Australian cohort, children from very remote areas and Aboriginal and Torres Strait Islander children were not well represented and care should be taken in extrapolating findings [52]. In the Québec cohort, children born in the Nord-du-Québec, Terres-Cries-de-la-Baie-James, Nunavik, or other Indian Reserves, about 2.1% of the population, were excluded [33]. The Rotterdam cohort included an urban population of children with multi-ethnic composition (68% Dutch/Western, 32% Non-Western ethnic background), for whom socio-economic disadvantage may be inequitably linked to relatively lower levels of health literacy and less favourable health behaviours [18].

## Public health implications

The current study findings align and provide further empirical evidence supporting recent recommendations from the WHO global strategy for oral health [5]. Key elements of this strategy (summarized in S2 Table) include the need for major system reforms to provide inclusive, accessible, and affordable oral health-care including the need to close financing gaps to align goals of universal primary health care coverage, and to establish data-driven evidence-informed policy for oral health (including the importance of tackling risk factors such as sugar consumption) and health care. While the WHO recommendations primarily focus on low- and middle-income countries, the current study is both timely and necessary to advance potential policy action towards equitable oral health and provision of care within high-income jurisdictions [5].

Inequalities in dental caries requires a broad public health approach that simultaneously focuses on the social determinants of health while more specifically considering population wide oral health and health care system solutions. Health care systems that offer and encourage free regular dental check-ups for all children and teenagers in public or private dental care are beneficial for public health. For example, in Southeast Sweden, the combination of universally available dental care (including for prevention), naturally fluoridated water, and low consumption of sugar seem to have resulted in a lower prevalence of dental caries for children. This suggests the combination of strategies that may be necessary to ameliorate the social gradient in childhood caries.

Similarly, it must be noted that fluoridated water is not available to all children in the four countries included in this study, despite its biologically demonstrated protective effects (and no demonstrated significant harm [23, 46]). Continued promotion of water fluoridation represents an important public health prevention target.

## Conclusions

Clear gradients of social inequalities for child dental caries exist across four high-income countries in this study. Variation in the prevalence of child dental caries and the extent of social inequalities across jurisdictions suggest that each may be responsive to oral health policy. The achievable combination of universal dental insurance and care (which includes adults/parents and children) and water fluoridation may be the necessary policy levers to achieve lower and more equitable rates of dental caries. To differentially benefit socio-economically disadvantaged groups of children, and sustain subsequent longer-term adult health benefits, it is timely to consider these universal and powerful public health measures in order to deliver on the aspirations of the WHO global oral health strategy.

## Supporting information

**S1 Table. Oral health policy and service mapping across countries.**
(DOCX)

**S2 Table. Recommendations for the new WHO global strategy for oral health supported by key findings from this study.**
(DOCX)

**S1 File. Supplementary reference list for S1 and S2 Tables.**
(DOCX)

## Acknowledgments

**Contributing Members of the EPOCH (Elucidating Pathways of Child Health inequalities) Collaborative Group include**: (PIs) Jennifer J. McGrath (PI, Concordia University, Canada; jennifer.mcgrath@concordia.ca), Louise Séguin (co-PI, Université de Montréal, Canada), Nicholas Spencer (co-PI, University of Warwick, UK), Kate Pickett (co-PI, University of York, UK), Hein Raat (co-PI, Erasmus MC, The Netherlands); (alphabetically) Yara Abu Awad (Concordia University, Canada), Pär Andersson White (Crown Princess Victoria Children's Hospital, Sweden), Guannan Bai (Erasmus MC, The Netherlands), Philippa Bird (Bradford Institute for Health Research, UK), Susan A. Clifford (The University of Melbourne, Australia), Åshild Faresjö (Linköping University, Sweden), Tomas Faresjö (Linköping University, Sweden), Kate L. Francis (Royal Children's Hospital, Australia), Lise Gauvin (Centre de recherche du CHUM & Université de Montréal, Canada), Sharon Goldfeld (The Royal Children's Hospital Melbourne, Australia), Jeremy D. Goldhaber-Fiebert (Stanford University, USA), Lea Kragt (Erasmus MC, The Netherlands), Johnny Ludvigsson (Linköping University, Sweden), Wolfgang Markham (University of Warwick, UK), Fiona K. Mensah (The University of Melbourne, Australia), Béatrice Nikiéma (formerly, Centre de recherche du CHUM, Canada), Elodie O'Connor (Royal Children's Hospital, Australia), Sue Woolfenden (University of New South Wales & Sydney Children's Hospital, Australia). Additional collaborators of the EPOCH (Elucidating Pathways of Child Health inequalities) Collaborative Group include: (alphabetically) Clare Blackburn (University of Warwick, UK), Sven Bremberg (Karolinska Institutet & National Institute of Public Health, Sweden), Anders Hjern (Centre for Health Equity Studies & Karolinska Institutet, Sweden), Jody Heymann (UCLA, USA), Lynn Kemp (Western Sydney University, Australia), Lisa Kakinami (Concordia University, Canada), Lucie Laflamme (Karolinska Institutet, Sweden), Johan Mackenbach (Erasmus MC, The Netherlands), Richard Massé (Ministère de la santé et des services sociaux, Gouvernement du Québec), Marie-France Raynault (Centre Hospitalier de l'Université de Montréal-CHUM,

Québec), Agatha van Meijeren—van Lunteren (Erasmus MC, The Netherlands), Paul Wise (Stanford University, USA), Junwen Yang-Huang (Erasmus MC, The Netherlands), Yueyue You (Erasmus MC, The Netherlands).

Sincere thanks to the dedicated EPOCH administrative staff, especially Sabrina Giovanniello & Julie Foisy (Research Coordinators), without whom this research would not be possible. We are grateful to all families who participated in the All Babies in Southeast Sweden (ABIS), Generation R Study (GenR), Longitudinal Study of Australian Children (LSAC), and Québec Longitudinal Study of Child Development (QLSCD). We thank Monsurul Hoq and Charlotte Molesworth (Clinical Epidemiology and Biostatistics Unit, MCRI, Australia) for their contributions to oral health and SEP data harmonization with the LSAC cohort.

## Author Contributions

**Conceptualization:** Sharon Goldfeld.

**Formal analysis:** Kate L. Francis, Johnny Ludvigsson, Tomas Faresjö, Beatrice Nikiema, Lise Gauvin, Yara Abu Awad, Lea Kragt.

**Funding acquisition:** Junwen Yang-Huang.

**Methodology:** Sharon Goldfeld, Kate L. Francis, Lea Kragt, Fiona K. Mensah.

**Project administration:** Elodie O'Connor.

**Supervision:** Sharon Goldfeld, Lea Kragt, Fiona K. Mensah.

**Writing – original draft:** Sharon Goldfeld, Kate L. Francis, Elodie O'Connor, Fiona K. Mensah.

**Writing – review & editing:** Sharon Goldfeld, Kate L. Francis, Elodie O'Connor, Johnny Ludvigsson, Tomas Faresjö, Beatrice Nikiema, Lise Gauvin, Junwen Yang-Huang, Yara Abu Awad, Jennifer J. McGrath, Jeremy D. Goldhaber-Fiebert, Åshild Faresjo, Hein Raat, Lea Kragt, Fiona K. Mensah.

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
