## [Decision Letter · Decision Letter 0]

8 Nov 2021

PONE-D-21-31950Oral health policy and comparative inequalities in child dental caries across four countries: An international birth cohort explorationPLOS ONE

Dear Dr. Goldfeld,

Thank you for submitting your manuscript to PLOS ONE. After careful consideration, we feel that it has merit but does not fully meet PLOS ONE’s publication criteria as it currently stands. Therefore, we invite you to submit a revised version of the manuscript that addresses the points raised during the review process.

We look forward to receiving your revised manuscript.

Kind regards,

Morenike Oluwatoyin Folayan, FWACS

Academic Editor

PLOS ONE

Journal Requirements:

3.  Please include your full ethics statement in the ‘Methods’ section of your manuscript file. In your statement, please include the full name of the IRB or ethics committee who approved or waived your study, as well as whether or not you obtained informed written or verbal consent. If consent was waived for your study, please include this information in your statement as well. .

4. Please provide additional details regarding participant consent. In the Methods section, please ensure that you have specified (1) whether consent was informed and (2) what type you obtained (for instance, written or verbal). If your study included minors, state whether you obtained consent from parents or guardians. If the need for consent was waived by the ethics committee, please include this information.

 [EPOCH was partly supported by Canadian Institutes of Health Research (J. McGrath OCO-79897, MOP-89886, MSH-95353; L. Séguin ROG-110537). 

Longitudinal Study of Australian Children (LSAC) was initiated and funded by Australian Government Department of Social Services, with additional funding from partner organizations Australian Institute of Family Studies (AIFS) and Australian Bureau of Statistics (ABS). This paper uses unit record data from Growing Up in Australia, the Longitudinal Study of Australian Children. The database of fluoride levels in water is maintained at the Australian Research Centre for Population Oral Health. The study was conducted in partnership with the Department of Social Services (DSS), the Australian Institute of Family Studies (AIFS) and the Australian Bureau of Statistics (ABS). The findings and views reported in this paper are those of the authors and should not be attributed to the DSS, the AIFS or the ABS. 

Generation R Study (GenR) was made possible by financial support from Erasmus Medical Center, Rotterdam; Erasmus University Rotterdam; Netherlands Organisation for Health Research and Development (ZonMw; additional grant received by V. Jaddoe, ZonMw 907.00303, 916.10159); Netherlands Organisation for Scientific Research (NWO); Ministry of Health, Welfare and Sport; and, Ministry of Youth and Families. GenR is conducted by Erasmus Medical Center in close collaboration with the School of Law and Faculty of Social Sciences of the Erasmus University Rotterdam, the Municipal Health Service Rotterdam area, Rotterdam, the Rotterdam Homecare Foundation, Rotterdam and the Stichting Trombosedienst & Artsenlaboratorium Rijnmond (STAR-MDC), Rotterdam; we gratefully acknowledge the contribution of children and parents, general practitioners, hospitals, midwives and pharmacies in Rotterdam.

Québec Longitudinal Study of Child Development (QLSCD) 1996-2014 cohort was principally funded by l’Institut de la statistique du Québec through partnership with Fondation Lucie et André Chagnon, Ministère de l’Éducation et de l’Enseignement supérieur, Ministère de la Santé et des Services sociaux, Ministère de la Famille, GRIP Research Unit on Children’s Psychosocial Maladjustment, QUALITY Cohort Collaborative Group, le Centre hospitalier  universitaire Sainte-Justine, Institut de recherche Robert-Sauvé en santé et en securité au travail, l’Institut de recherche en santé publique de l’Université de Montréal, Centre de recherche du Centre hospitalier de l’Université de Montréal (CRCHUM), Fonds de recherche du Québec Santé (FRQS), Fonds de recherche du Québec Sociéte et culture (FRQSC), Social Sciences and Humanities Research Council (SSHRC), and Canadian Institutes of Health Research (MOP-123079, HDF-70335). The paper used unit record data from the QLSCD (ELDEQ – Enquête longitudinale des enfants du Québec). Data for the QLSCD were collected by the Institut de la Statistique du Québec, Direction des enquêtes longitudinales et sociales.

ABIS (Alla Barn i Sydöstra Sverige; All Babies in Southeast Sweden) and this research were supported in part by the County Council of Ostergotland, Swedish Research Council (K2005-72X-11242-11A and K2008-69X-20826-01-4), the Swedish Child Diabetes Foundation (Barndiabetesfonden), Juvenile Diabetes Research Foundation, Wallenberg Foundation (K 98-99D-12813-01A), Medical Research Council of Southeast Sweden (FORSS), the Swedish Council for Working Life and Social Research (FAS2004–1775), and Ostgota Brandstodsbolag. Johnny Ludvisson founded the ABIS Cohort.

Sharon Goldfeld is supported by Australian National Health and Medical Research Council (NHMRC) Practitioner Fellowship 1155290. Fiona Mensah was supported by NHMRC Career Development Fellowship 1111160. Research at the Murdoch Children’s Research Institute is supported by the Victorian Government’s Operational Infrastructure Program.]

6. We note that you have indicated that data from this study are available upon request. PLOS only allows data to be available upon request if there are legal or ethical restrictions on sharing data publicly. For more information on unacceptable data access restrictions, please see http://journals.plos.org/plosone/s/data-availability#loc-unacceptable-data-access-restrictions. 

7. One of the noted authors is a group or consortium [EPOCH Collaborative Group]. In addition to naming the author group, please list the individual authors and affiliations within this group in the acknowledgments section of your manuscript. Please also indicate clearly a lead author for this group along with a contact email address.

Reviewers' comments:

Reviewer's Responses to Questions

**Comments to the Author**

1. Is the manuscript technically sound, and do the data support the conclusions?

Reviewer #1: Yes

Reviewer #2: Partly

2. Has the statistical analysis been performed appropriately and rigorously? 

Reviewer #1: Yes

Reviewer #2: Yes

3. Have the authors made all data underlying the findings in their manuscript fully available?

Reviewer #1: Yes

Reviewer #2: Yes

4. Is the manuscript presented in an intelligible fashion and written in standard English?

Reviewer #1: Yes

Reviewer #2: Yes

5. Review Comments to the Author

Reviewer #1: The study discussed an important topic and its results are valuable to scientific community.

Introduction:

The reviewer suggests some changes

*page#4, Line 99 : Replace 'lower-income' by 'Low-income countries'. Also change 'higher-income' to 'high-income'

*The aim of the study (objectives) should be written in a clear way.

*The rationale of the study & Null-hypothesis should be clearly stated

*Furthermore, the authors massively used adjectives (We/Our) which sometimes weakening the scientific structure, hence, the reviewer suggest to revise these sentences

Methodology :

*Table text citation should be done in a proper way (e.g Table 1, Table 2 , ......etc)

Conclusions :

It looks like results , hence the reviewer suggests to revise.

Overall comment about the manuscript language: It seems that this manuscript was written by native English speakers, however , the reviewer highly recommends revising the manuscript text by an expert academic writer to fulfill basic requirements of scientific writing

Reviewer #2: The current paper investigated inequalities in the prevalence of dental caries across four countries using birth cohorts. As the authors highlighted, dental caries although preventable still a burden especially among underserved and disadvantaged populations. Therefore there is an urgent need to investigate these inequalities and policies to address them. The paper is well written and the multi-centre methodology allows the reader to have a holistic view of different countries approaches. However, here are some points/concerns that the authors need to consider:

_ Title: the "Oral health policy" in the title may not be appropriate as this was not investigated nor was it part of the analysis but rather discussed in the context of the results. It might be beneficial if these policies were part of the variables investigated.

_ Introduction: The authors may consider summarizing the sections related to caries risk factors and elaborate on the caries prevalence in the four countries. Any previous studies on caries prevalence/risk factors? Also it is important to give an idea about current preventive programs/measures and policies, similar to what was mentioned about Australia.

- Materials and Methods:

1. Please write the exact period of the study for each country.

2. From where was the participants recruited? hospitals or community centres?

3. What were the inclusion and exclusion criteria?

4. The participants were investigated/examined at single time or multiple time slots? please clarify.

5. For those who were examined, who conducted the examination? please provide details on the examination process and calibration.

6. For those whose parents reported on their caries: was that through an interview or questionnaire?

7. How was caries risk factors investigated ? If a questionnaire was used, why oral hygiene habits, dental visits or exposure to preventive measures investigated?

8. "Oral health provision is thus explored at a policy level by country/region" I still don't see how was this assessed in relation to study outcome.

9. Please report on consent from patients.

10. Any information about children/mothers' medical health?

-Results:

1. Please report on total number of participants for each country. Were there any loss of participants? It would be beneficial if the authors use a chart to clarify the recruitment process over time.

2. The authors may consider re-arranging this section either by country and report findings under each or by variables investigated and report each country separately for more clarity.

-Discussion:

- Table 3: It is uncommon using tables as part of the discussion, I believe instead it would be better if these points/recommendations are discussed in the context of the the evident provided and the study results.

- Abstract:

Needs to be re-written in light of the concerns raised.

- Please use STROBE checklist when revising the manuscript.

6. PLOS authors have the option to publish the peer review history of their article (what does this mean?). If published, this will include your full peer review and any attached files.

Reviewer #1: **Yes: **Hamdi Hosni Hamama

Reviewer #2: **Yes: **Balgis Osman Gaffar

---

## [Author Response · Author response to Decision Letter 0]

7 Mar 2022

We have amended our Data Availability statement as requested. It now reads:

Data underlying the results presented in this EPOCH study are available from the primary data sources. Data from LSAC is available in a public, open-access repository (https://growingupinaustralia.gov.au/data-anddocumentation). Data from GenR are available to request from (https://generationr.nl/researchers/); authors do not have permission to share their data. Data from ABIS are available to request from (http://www.abis-studien.se); authors do not have permission to share their data. Data from QLSCD is available to request from (https://www.maelstrom-research.org/mica/individualstudy/qlscd#); authors do not have permission to share their data. Other researchers can access these data in the same manner as the authors, as there were no special access privileges granted.

---

## [Decision Letter · Decision Letter 1]

11 May 2022

Comparative inequalities in child dental caries across four countries: Examination of international birth cohorts and implications for oral health policy

PONE-D-21-31950R1,

We’re pleased to inform you that your manuscript has been judged scientifically suitable for publication and will be formally accepted for publication once it meets all outstanding technical requirements.

Kind regards,

Morenike Oluwatoyin Folayan, FWACS

Academic Editor

PLOS ONE

Additional Editor Comments (optional):

Thanks for submitting your comprehensive response to the manuscript

Reviewers' comments:

Reviewer's Responses to Questions

**Comments to the Author**

1. If the authors have adequately addressed your comments raised in a previous round of review and you feel that this manuscript is now acceptable for publication, you may indicate that here to bypass the “Comments to the Author” section, enter your conflict of interest statement in the “Confidential to Editor” section, and submit your "Accept" recommendation.

Reviewer #2: All comments have been addressed

2. Is the manuscript technically sound, and do the data support the conclusions?

Reviewer #2: Yes

3. Has the statistical analysis been performed appropriately and rigorously? 

Reviewer #2: Yes

4. Have the authors made all data underlying the findings in their manuscript fully available?

Reviewer #2: Yes

5. Is the manuscript presented in an intelligible fashion and written in standard English?

Reviewer #2: Yes

6. Review Comments to the Author

Reviewer #2: The authors have addressed all comments efficiently. The manuscript can be accepted in its current state.

7. PLOS authors have the option to publish the peer review history of their article (what does this mean?). If published, this will include your full peer review and any attached files.

Reviewer #2: **Yes: **Balgis Gaffar